# Stem Cell Factors BAM1 and WOX1 Suppressing Longitudinal Cell Division of Margin Cells Evoked by Low-Concentration Auxin in Young Cotyledon of Arabidopsis

**DOI:** 10.3390/ijms26104724

**Published:** 2025-05-15

**Authors:** Yuli Jiang, Jian Liang, Chunyan Wang, Li Tan, Yoji Kawano, Shingo Nagawa

**Affiliations:** 1Institute for Translational Brain Reaearch, Fudan University, Shanghai 200032, China; 2Center for Excellence in Molecular Plant Science, Chinese Academy of Sciences, Shanghai 200032, China; leungjian@outlook.com (J.L.); wangcy@dhu.edu.cn (C.W.); tanli@bsbii.cn (L.T.); shinnagawa@hotmail.com (S.N.); 3Institute of Plant Science and Resources, Okayama University, Okayama 710-0046, Japan; yoji.kawano@okayama-u.ac.jp

**Keywords:** BAM1, WOX1, margin cells, auxin

## Abstract

Highly differentiated tissues and organs play essential biological functions in multicellular organisms. Coordination of organ developmental process with tissue differentiation is necessary to achieve proper development of mature organs, but mechanisms for such coordination are not well understood. We used cotyledon margin cells from Arabidopsis plant as a new model system to investigate cell elongation and cell division during organ growth and found that margin cells endured a developmental phase transition from the “elongation” phase to the “elongation and division” phase at the early stage in germinating seedlings. We also discovered that the stem cell factors BARELY ANY MERISTEM 1 (BAM1) and WUSCHEL-related homeobox1 (WOX1) are involved in the regulation of margin cell developmental phase transition. Furthermore, exogenous auxin treatment (1 nanomolar,nM) promotes cell division, especially longitudinal cell division. This promotion of cell division did not occur in *bam1* and *wox1* mutants. Based on these findings, we hypothesized a new “moderate auxin concentration” model which emphasizes that a moderate auxin concentration is the key to triggering the developmental transition of meristematic cells.

## 1. Introduction

Plant leaf epidermis consists of several differentiated cell types, such as stomatal guard cells for gas exchange and trichomes that may serve protective roles for herbicides. Pavement cells (PCs) are the most frequent cell types in the leaf epidermis in the model plant Arabidopsis [1,2]. They form a characteristic intercalating structure with multiple lobes and indentations. PCs in the adaxial side of leaves show distinctively different shapes from PCs in the abaxial sides, and those 2 types of cells are separated by a few layers of marginal cells with elongated cylindrical shapes [3]. Marginal cells have roles beyond the physical distinction of adaxial/abaxial cells since they are suggested to transport the essential hormone auxin [4,5]. Moreover, a recent report showed that marginal cells have meristematic activity specifically at the early stage of cotyledon/leaf development, which only maintains for a few days after germination of cotyledon or emergence of leaves [6]. WUSCEL (WUS) and related homeobox transcription factors are regulators of stem cell activity in different tissues such as shoot apical meristem (SAM), root apical meristem (RAM) and vascular cambium [7]. Among *WUSCHEL-RELATED HOMEOBOX (WOX)* genes, *WOX1* and *PRESSED FLOWER (PRS)/WOX3* are involved in the development of marginal cells. Plants lacking the function of those genes have multiple defects in leaves and floral organs, most notably defects in blade outgrowth [8]. The double mutant is reported to lose marginal cells in leaves as well as sepals [9]. Both WOX1 and PRS are expressed in the middle domain of leaf primordium and marginal cells [10], supporting their role in maintaining stem cell activity of those cells. Further support for their roles in maintaining stem cell activity of marginal cells came from studies on orthologous genes in other plant species that have similar functions [11,12]. Adaxial-expressed MONOPTEROS (MP), abaxial-enriched auxin and abaxial-expressed auxin response factor (ARF) repressors together act as positional cues for defining the *WOX* expression in margin cells. MP regulates the expression of *WOX* genes by directly binding to the *WOX1* and *PRS* promoters. Furthermore, the ARF2, ARF3, and ARF4 repressors suppress *WOX1* and *PRS* expression, also through direct binding [10]. This work suggested that the nuclear auxin signal pathway is involved in WOX-regulated leaf margin development. However, the molecular mechanisms underlying the acquisition of meristematic activity as well as the developmental phase shift to lose such activity in marginal cells are unknown.

Phosphorylation of proteins has important biological significance not only by altering the properties of the phosphorylated protein itself but also by inducing signaling cascades to enable dynamic regulation of downstream processes. In plants, receptor-like kinases (RLKs)-mediated signaling plays key roles in multiple developmental events [13]. Small peptides as well as other small molecule ligands such as plant hormones are perceived by RLKs for activating downstream signaling [14]. Signaling mediated by the CLAVATA3 (CLV3) peptide-CLAVATA1 (CLV1) RLK pair is involved in stem cell maintenance in the SAM and is one of the most extensively characterized examples of small peptide-mediated signaling in plants [15]. CLAVATA3/ESR (CLE) peptides act as mobile ligands in apoplast to activate signaling in neighboring cells [16], and they typically are involved in the regulation of stem cells in plants [17]. Direct binding of CLV3 as well as other CLE peptides to several closely related RLKs including CLV1, BAM1 and BARELY ANY MERISTEM2 (BAM2) has been detected [18], and the signaling module is proposed to restrict meristematic activity of SAM stem cells by repressing WUS expression in the central zone of the SAM [19]. Hormonal signaling is believed to cross-talk with WUS-mediated signaling in both upstream and downstream of WUS [20,21,22]. Although CLV1-related RLKs are proposed to have a function beyond SAM based on severe abnormalities induced by the loss of function of multiple *BAM* genes [23], the exact function of these proteins in other tissues as well as their relationship to WOXs has not been investigated in detail yet.

In the root tip, the auxin maximum is considered as a polarity signal for the cell layer patterning [24,25]. Besides auxin maximum, low-concentration auxin also plays a critical role in plant development. For example, during leaf adaxial-abaxial patterning, it has been shown that a transient low auxin zone in the adaxial domain of early leaf primordia worked as a signal to promote leaf adaxial-abaxial patterning [26]. The formation of a local auxin minimum is necessary for the specification of the valve margin separation layer where the Arabidopsis fruit opening takes place [27]. It is reported that auxin minimum triggers the developmental switch from cell division to cell differentiation in the Arabidopsis root [28].

In this work, we used cotyledon margin cells as a new system to study the meristematic activity of plant cells through analysis of cell elongation and cell division. In the root, the transition of the cell division stage to the cell elongation stage occurs successively along the root growth axis determined by the position of cells and thus regulated by spatial positional cues. In contrast, switching of the cell division phase to cell division with the cell elongation phase in margin cells seems to be regulated in a temporal and spatial manner. By utilizing this beneficial feature of margin cells for analysis, we demonstrated that stem cell factors BAM1 and WOX1 are involved in the suppression of cell division of margin cells evoked by low-concentration auxin, which may connect auxin and stem cell factors in regulating the meristematic activity of stem cells in plants.

## 2. Results

### 2.1. Cell Elongation of Margin Cells Precedes Cell Division in Wild-Type Cotyledons During Germination

Epidermal cells located at the marginal region in cotyledons and leaves play important roles in leaf shape formation [29,30] and plants adaptation to the environment [2]. Although margin cells have been recognized as specialized cells with distinct shapes compared to other cell types in the leaf epidermis such as pavement cells or stomatal lineage cells, the developmental process of margin cells has not been characterized in detail [6,31]. There is no explicit definition of margin cells in Arabidopsis cotyledon. In this study, we only focused attention on the rectangle cells located at the outermost adaxial side and used them to represent margin cells while the small round cells beside the rectangle cells and the rectangle cells located in the abaxial side were not counted. We used the mPS-PI staining method [32] to visualize cell walls and measured cell number and longitudinal cell length. Since the cotyledon is plane-symmetric, we analyzed one side of the cotyledon. Through counting the rectangle cells located at the outermost region of the adaxial epidermis of wild-type Col-0 (Col) plants at 1 dag (day after growth condition) to 3 dag (Figure 1A–C), we found that the cell number of margin cells of 1 dag cotyledons had little variation ranging from 13 cells to 15 cells. The cell number of margin cells at 2 dag did not show significant change compared to 1 dag (Figure 1D). These results indicated that margin cells did not endure cell division at 1 dag and 2 dag. In contrast, we found that the cell number increased at 3 dag (Figure 1D). Then we measured the longitudinal cell length from the top of the margin cell to the bottom of the margin cell. The quantification of cell length in 1 dag, 2 dag and 3 dag showed that margin cells already increased in cell length at 2 dag, and thus they endured cell elongation without division before 2 dag (Figure 1E).

Thus, we concluded that the cell elongation of margin cells precedes the cell division in developing wild-type cotyledons, and the developmental phase transition from the “elongation” phase to the “elongation and division” phase occurs in cotyledons right after germination.

### 2.2. BAM1 Regulates Margin Cell Developmental Phase Transition

Our data showed that the cell elongation of margin cells precedes the cell division in developing wild-type cotyledons, and we hypothesized that there is a signaling mechanism to regulate such developmental phase transition. We have previously found that *bam1-3* mutant (SALK_015302; Appendix A) has increased the number of margin cells at 2 dag [33]. Thus, we decided to check if *bam1-3* shows abnormality in the margin cell developmental phase transition (Figure 2A–E). At 1 dag, the cell number of margin cells in both Col and *bam1-3* cotyledons showed little variation ranging from 13 cells to 15 cells. However, unlike Col cotyledons, the cell number of margin cells of 2 dag cotyledons in *bam1-3* increased up to 17 cells indicating that the cell division occurred between 1 dag to 2 dag (Figure 2A,B,D). In contrast, unlike Col plants, the cell length of *bam1-3* did not increase from 1 dag to 2 dag (Figure 2E). Similar to our previous analysis [33], the cell number of palisade cells (Appendix A) and cotyledon size (Appendix A) of Col and *bam1-3* showed no significant difference at 2 dag, suggesting that the defect of *bam1-3* is specific to margin cells. Our results show that margin cells elongated before 2 dag and then started to divide along with the growth of cotyledon. Suppression of margin cell division before 2 dag is dependent on BAM1.

To test whether other BAM protein members, BAM2 and BAM3, are involved in regulating margin cell developmental phase transition, we observed margin cell development of *bam1 bam2 bam3* triple mutant and found that *bam1 bam2 bam3* displayed a margin cell developmental phase transition defect as *bam1-3*, neither reinforcing nor alleviating the margin cell developmental phase transition defect of *bam1-3* mutant (Figure 2C–E and Appendix A). This data suggested that there is no genetic redundancy between BAM1 and BAM2/BAM3 in regulating margin cell developmental phase transition.

Next, to further confirm the increase in cell division in *bam1-3*, we applied a pulse-chase strategy for EdU labeling assay [34] to quantify cell division frequency of margin cells of 1 dag, 1.5 dag and 2 dag in Col and *bam1-3* mutant. The EdU assay can monitor the cell division that occurs during the EdU treatment. Pulse-chasing allowed us to monitor the cell division orientation by identifying daughter cell pairs. We found that *bam1-3* displayed more longitudinal cell divisions than Col, from 30.8% to 15.2% of 1 dag and from 43.2% to 28.4% of 1.5 dag, respectively (Table 1). These data confirmed that longitudinal cell division is increased during 1 dag to 2 dag in *bam1-3* compared to Col.

### 2.3. Margin Cells of bam1-3 Mutants Show Abnormal Auxin Response Monitored by an Auxin Response Marker

Auxin regulates the timing of cell division or cell differentiation in different tissues [35,36,37]. Since the developmental phase transition of margin cells is altered in *bam1-3* mutant, it is necessary to check whether the auxin response changed in *bam1-3* mutant. We checked the GFP signal in pDR5::GFP-ER and found that the GFP intensity increased from 1 dag to 2 dag with maximum intensity mainly around the tip area (Figure 3A). The maximum GFP intensity displayed in the cotyledon tip in *bam1-3* at 1.5 dag demonstrates that the auxin accumulation at the tip is not affected in *bam1-3* (Figure 3B). However, the GFP signal is much higher in margin cells compared to other epidermal cells in *bam1-3* at 1.5 dag and 2 dag (Figure 3B). In Col cotyledons, the margin regions show only a slight increase in GFP intensity compared to other regions. However, we detected a significant increase in GFP intensity in the margin region of *bam1-3* cotyledons (Figure 3C). These results suggested that margin cells may endure higher auxin response compared to other cell types in *bam1-3.*

### 2.4. WOX1 Coordinately Regulates Margin Cell Development with BAM1

Previous reports showed that WUS-like transcription factors WOX1 and PRS are involved in the development of marginal cells in various shoot organs [38,39,40], and *wox1 prs* lacks leaf margin cells [9]. We checked the cotyledon of *wox1 prs* and found that the long, rectangular margin cells, were also absent in *wox1 prs* cotyledons (Figure 4G). Then, we checked the single loss of function alleles for *WOX1* and *PRS* on developmental defects in margin cells. Unlike margin cells of Col cotyledons which did not divide before 2 dag, margin cells of the *wox1-101* cotyledons divided from 1 dag similar to *bam1-3* (Figure 4B,C). In contrast, *prs-2* cotyledons did not show alteration in margin cell development (Figure 4D). Neither *wox1 bam1-3* nor *prs-2 bam*1-3 showed enhancement of margin cell developmental defects from defects observed for *bam1-3* (Figure 4E,F). We crossed *bam1-3* with *wox1 prs*, and found that the *bam1 wox1 prs* triple mutant lacks margin cells as in the *wox1 prs* mutant (Figure 4H). Quantification of cell division and cell elongation in mutants confirmed that bam1-3 and wox1-101 show similar alterations in the developmental phase transition of margin cells (Figure 4I,J). However, in *wox1-101* and *wox1 bam1*, the cell number of margin cells increased up to 15 and 16, respectively, at 1 dag (Figure 4I). Given that in both Col and *bam1-3* the cell number of margin cells at 1 dag is constant around 14, the results suggested that WOX1 may regulate margin cell development in an early stage before 1 dag.

Next, in order to detect early developmental defects before germination, we checked mature embryos of Col, *bam1-3, wox1-101* and *wox1 bam1* (Figure 5A–D). We found that the margin cell number of mature embryos in *bam1-3* and *wox1-101* are similar to Col. The cell number of mature embryo margin cells in *wox1 bam1* is increased slightly to around 15 (Figure 5E) and the cell length of this mutant is decreased slightly (Figure 5F), suggesting that the BAM1 has synergistic function with WOX1 in regulating margin cell numbers in embryo stage.

The pulse-chase EdU labeling assay showed that at 1 dag the frequency of longitudinal cell divisions in *wox1-101* and *wox1 bam1* increased, from 15.2% to 35.2% and 15.2% to 26.8% respectively. However, at 2 dag, the frequency of longitudinal cell divisions in *wox1-101* and *wox1 bam1* decreased, from 14.4% to 4.4% and 14.4% to 6.8% respectively (Table 1). Thus, the synergistic effect of combining *wox1-101* and *bam1-3* is limited to the embryo development stage, while it shows no enhancement of abnormality in post-embryonic development.

To check the expression pattern of BAM1 and WOX1 in young cotyledons, we constructed the promoter reporter line for *BAM1* and *WOX1* which contained YFP fused with nucleus localization signal downstream of the promoter sequence. Consistent with the defects at the germinating cotyledons, the transgenic line *proBAM1::YFP-NLS* started to show a YFP signal from the middle region of cotyledon at 1 dag and the signal expanded to appear on cells over the whole cotyledon after 2 dag (Appendix A). The transgenic line *proWOX1::YFP-NLS* started to show a YFP signal from the margin cells of cotyledons at 1 dag and the signal was restricted to the margin region till a later stage (Appendix A). These results suggested that the margin cell development in cotyledon is regulated by BAM1 and WOX1 at different stages. WOX1 works from the mature embryo to 2 dag while BAM1 may act at a relatively later stage. However, BAM1 seems to play roles in margin cell development in the embryo stage with WOX1 which may be hindered by genetic redundancy with other RLKs.

### 2.5. Low-Concentration Auxin Suppresses Longitudinal Cell Division of Margin Cells

Next, to gain further insights into the effect of auxin on margin cell development, we examined whether different concentrations of auxin has a differential effect on cell division of margin cells in Col by applying an EdU labeling assay to monitor cell division. Seedlings were divided into 5 groups and cultured in 1/2 MS with 0.1 nM NAA, 1 nM NAA, 10 nM NAA and EtOH as control, respectively, before an EdU staining assay at 1.5 dag. The results demonstrated that with 1 nM NAA treatment, the frequency of transversal and longitudinal cell divisions increased, from 25.3% to 30.7% and from 24.0% to 42.7%, respectively (Table 2). With 100 nM NAA treatment, the frequency of transversal and longitudinal cell divisions decreased, from 25.3% to 10.0% and from 24.0% to 11.3%, respectively (Table 2). These results suggested that 1 nM NAA treatment accelerates longitudinal cell division while 100 nM NAA depresses both transversal and longitudinal cell division.

Then we tested the potential negative contribution of BAM1 and WOX1 in the low-concentration auxin response since *bam1-3* and *wox1-1* showed similar alteration of cell division frequency to the low-concentration auxin treatment. When we applied pulse-chase EdU labeling assay with 1 nM NAA treatment in *bam1-3* and *wox1-101* at 1.5 dag, we found that neither *bam1-3* nor *wox1-101* showed increased cell division frequency compared to Col (Table 3). All together, these data suggest that BAM1 and WOX1 act as negative regulators of longitudinal cell division induced by low-concentration auxin treatment.

## 3. Discussion

Our previous data showed margin cells’ developmental phase transition occurs at earlier timing in both *bam1-3* and *wox1-101*. Phenotype analysis of mature embryo margin cells in *wox1-101*, *bam1-3* and *bam1 wox1* demonstrated that BAM1 and WOX1 work synergistically to control embryo margin cells’ development. Unlike the opposite phenotype of *clv1* and *wus* exhibited in the CLV3-WUS feedback loop of controlling SAM maintenance, *wox1-101* and *bam1-3* seem to have an abnormal phenotype in a similar direction. Increased division of stem cells may result in loss of stem cell activity, which in turn leads to the consumption of proliferating cells in the meristem. Detailed analysis of mutants with a weaker phenotype due to the reduction of *WUS* genes and genetic analysis among multiple mutants in the CLV-WUS pathway using such weak mutants may reveal whether this idea is correct or not.

As Figure 3A shows, the gradient of auxin response is formed over the cotyledon surface with a pronounced accumulation of signal along margin cells by 2 dag. Thus, it is reasonable to assume that increased auxin evokes the cell division after 2 dag. Interestingly, exogenously applied low-concentration NAA promotes cell division while high-concentration NAA application inhibits cell division. These data suggest that a moderate auxin concentration is necessary for margin cells to transit from the “elongation” phase to the “elongation and division” phase, and a higher or lower concentration of auxin out of that range keeps cells to be elongated.

Auxin concentration is controlled by auxin biosynthesis, auxin transport and auxin degradation. Since our data were obtained from exogenously applied auxin, the exact concentration range of auxin to induce margin cell division is currently unclear. Further experiments such as characterizing the development of margin cells in auxin synthesis mutant lines or testing the effect of blocking auxin transportation or inhibiting auxin degradation will be conducted in the future. Different from a previous hypothesis that the auxin maximum or auxin minimum triggers the developmental switch [41], our work suggested that an appropriate moderate auxin concentration triggers the transition of margin cells from the cell elongation to the cell division. Thus, we would like to introduce a new “moderate auxin concentration model”, which proposes the existence of mechanisms to achieve switching of the developmental program specifically by a certain auxin concentration range.

The major differences between the auxin maximum or minimum model and the moderate auxin concentration model are summarized in Figure 6A. The auxin maximum or minimum model emphasizes that achieving high or low auxin concentration works as a trigger to developmental regulation. The moderate auxin concentration model proposes that there is a moderate range for auxin to trigger developmental regulation. Too high or too low auxin concentration will not trigger the relevant developmental regulation (Figure 6B). However, the model was drawn mainly by exogenous NAA treatment and it is worthwhile to analyze the internal auxin signaling in the margin cells by 1 nM and 100 nM NAA treatment to conclude the model in the future.

It is possible that the reported auxin minimum is actually a moderate auxin concentration instead of minimum auxin concentration since the auxin markers used in those studies, DR5-GFP or DII-VENUS, are both based on TIR-mediated auxin response expected to reflect only a certain range of high auxin concentration. To confirm this idea, a new auxin marker to detect low auxin concentration at a few nM range needs to be developed and investigated in each developmental context where auxin minimum is reported to be playing an important role. Experiments using auxin-inducible lines and auxin markers in auxin biosynthesis mutants to control auxin concentration artificially should be conducted.

Plant development is complicated, and auxin concentration is not the only factor determine margin cells’ fate. It is reported that cytokinin and brassinosteroid can affect cell elongation and cell division together with or without auxin as well [42,43]. Whether the two hormones also participate in the margin cell development and whether they have cross-talk with auxin need to be clarified in the future. Moreover, cell division is a result of the cell cycle. Whether the cell division increase triggered by moderate auxin concentration changes the expression of some key regulators in the cell cycle, for example, translation elongation factor 2 (EF2) needs to be investigated in future studies [44].

## 4. Materials and Methods

### 4.1. Plant Materials and Growth Conditions

All Arabidopsis mutants and transgenic plants used in this work are in the Col-0 background. The *bam1-3* mutant is described in DeYoung et al., 2006 [45]. The *wox1prs2* double mutant is described in Nakata et al., 2012 [9]. The seeds are kindly offered by Prof. Kiyotaka Okada. *wox1-101* and *prs-2* single mutant is generated through crossing. *bam1 DR5-GFP* is generated through crossing. *bam1 bam2 bam3* is from Dr. Rosa Lozano-Duran. The promoter sequences of *BAM1* and *WOX1* were amplified by PCR and cloned into a *pBGYN* vector described in Kubo et cl, 2005 [46].

Arabidopsis seeds were surface-sterilized in 10% NaClO and 0.1% triton for 30 s and rinsed with sterile water for 1 min, repeating 4 times. Then seeds were placed at 4 °C for 4 days. Afterward, we placed the seeds on 1/2 MS nutrient medium (Murashige and Skoog salts with 1% *w/v* of sucrose, PH = 5.8, adjusted by KOH) plates with 0.8% (*w/v*) agar. The plates were then transferred to growth conditions with 16 h light/8 h dark cycles, 22 °C and relative humidity of 40%.

### 4.2. RNA Extraction and qRT-PCR

Total RNA was extracted from Col-0 and *bam1-3* seedlings at 1.5 dag using ‘RNeasy Plant Mini Kit’ supplied by QIAGEN; 1 µg total RNA of each sample was used for synthesizing cDNA using ‘TransScript II One-Step gDNA Removal and cDNA Synthesis SuperMix’ from Transgene company, Beijing, China. The primers used for quantitative RT-PCR were designed using the Primer Premier software 5.0. The sequences are shown in Appendix A.

### 4.3. mPS-PI Staining

Whole seedlings were fixed in 2 mL of an aged solution of ethanol:acetic acid (3:1) (stored for a minimum of 4 months at 4 °C), and then put under vacuum for 30 min and left on a shaker at 4 °C for 2 days. Then, gradient dehydration was carried out using alcohol. Subsequently, the samples were treated with 0.01% amylase at 37 °C overnight. After that, they were stained with pseudo-schiff containing 0.01% PI, washed several times with ddH2O and finally stored in chloral hydrate for observation [32].

### 4.4. EdU Labeling Assay

Whole seedlings without any damage were cultured in 1 mL 1/2 Murashige and Skoog Medium (MS) liquid medium containing 1% (*w/v*) sucrose in growth conditions with 10 μM EdU for 45 min followed by 8 h MS liquid medium culture without EdU. Then, the 1/2 MS liquid medium was replaced with FAA (50% ethyl alcohol, 2.5% glacial acetic acid and 2.5% formalin), and left overnight at 4 °C, without vacuum. After that, seedlings were subjected to a cocktail treatment and eluted twice with ddH_2_O. Then, the seedlings were observed under Zeiss Axio Imager Z2 (Carl Zeiss, Oberkochen, Germany) microscope [34].

### 4.5. Microscopy

Leica TCS SP8 confocal microscope was used for acquiring the images. PI was excited with a 561 nm solid-state laser, 10% laser power and emission was detected at bandwidths of 570–620 nm. Z-stack was applied for each sample and the step size was set as 2 μm. For GFP, excitation was achieved by 488 nm from Argon laser, 40% laser power and emission was set at 500–550 nm. For DAPI, excitation was achieved by 405 nm solid-state laser, 10% laser power and emission was set at 415–450 nm.

### 4.6. NAA Treatment

Sterilized seeds were added into the 1/2 MS liquid media with different concentrations of NAA or the corresponding amount of EtOH. Then 1 mL liquid was mixed with the 1/2 MS liquid media, and the seeds and the chemical were added to the 24-well plate individually. The plate was rotated at 100 rounds per minute in the greenhouse and seedlings were grown for 2 days.

## Figures and Tables

**Figure 1 ijms-26-04724-f001:**
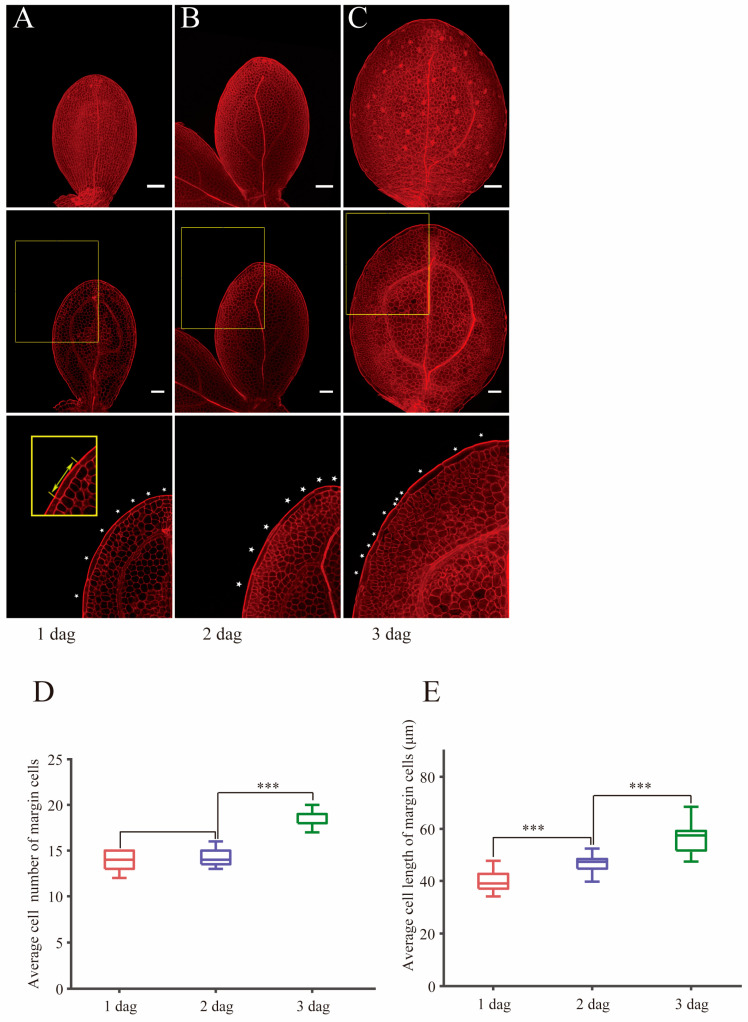
Cell elongation of margin cells preceding cell division in wild-type cotyledons during germination process. (**A**–**C**) Images of cotyledons with mPS-PI staining at 1 dag (**A**), 2 dag (**B**) and 3 dag (**C**) in Col. Maximum projection images of Z-stacks are shown in upper panel. Single slice of optimal sections of margin cells are shown in middle panel. Bottom panels show magnified images of middle panels (outlined in yellow boxes). White stars indicate margin cells. Bars = 50 μm. (**D**,**E**) Quantitative analysis of average cell number (**D**) and average cell length (**E**) of margin cells in (**A**–**C**). Cell lengths are measured by drawing a straight line in the longitudinal axis along a cell as indicated in the bottom left panel in (**A**). Data are presented as mean ± SD. Student’s *t*-tests were performed based on the differences of 2 dag to 1 dag and 3 dag to 2 dag (*** *p* < 0.001, *n* = 16).

**Figure 2 ijms-26-04724-f002:**
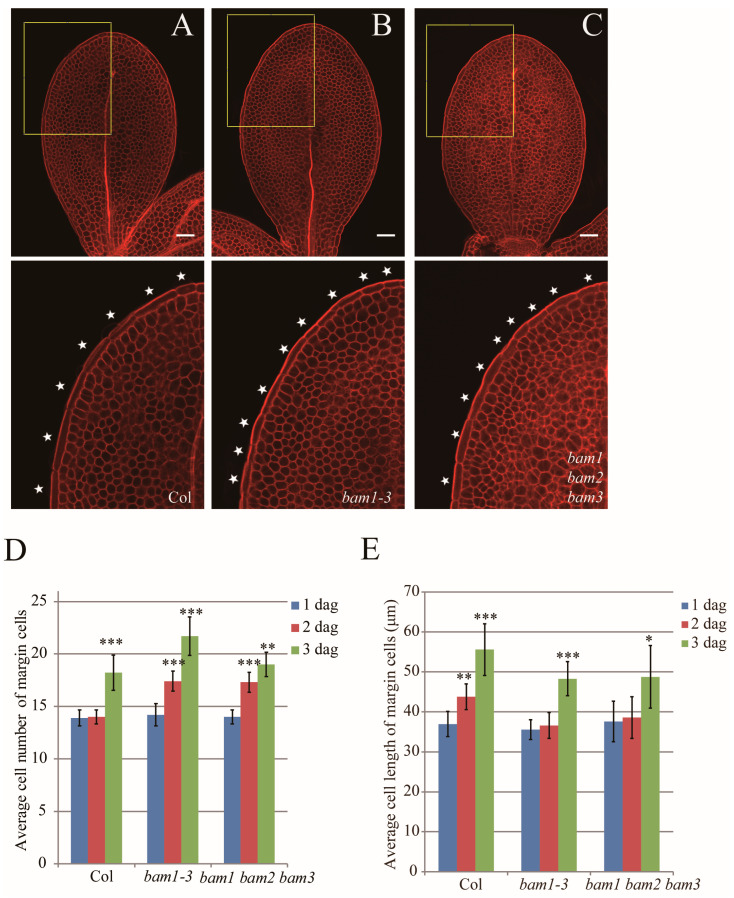
BAM1 regulating margin cell developmental phase transition. (**A**–**C**) mPS-PI stained 2 dag cotyledons of Col (**A**), *bam1-3* (**B**) and *bam1, 2, 3* (**C**). Single optimal section of margin cells is shown in upper panel. Bottom panels show magnified images of upper panels (outlined in yellow boxes). White stars indicate margin cells. Bars = 50 μm. (**D**,**E**) Quantitative analysis of average cell number (**D**) and average cell length (**E**) of margin cells in Col, *bam1-3* and *bam1, 2, 3*. Data are presented as mean ± SD. Student’s *t*-test was performed to assess significance of differences between 2 dag to 1 dag and 3 dag to 2 dag in each group. The difference between each mutant line and Col was determined by two-way ANOVA. (*, ** and *** denote *p* < 0.05, *p* < 0.01 and *p* < 0.001, respectively. *n* = 8).

**Figure 3 ijms-26-04724-f003:**
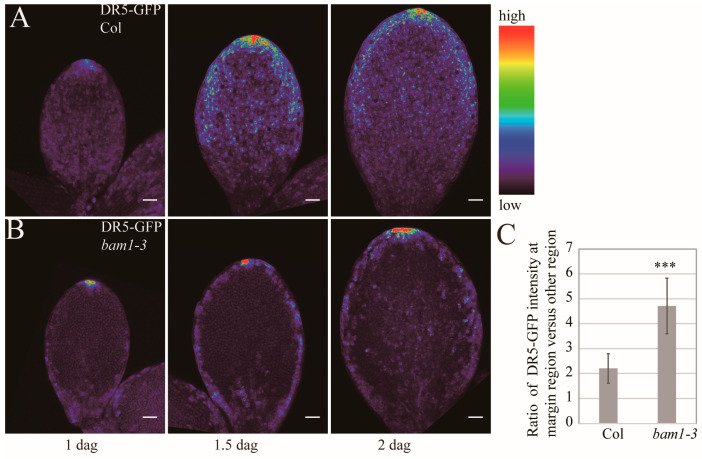
Margin cells of *bam1-3* mutant show abnormal accumulation of GFP signal from *DR5-GFP.* (**A**,**B**) DR5-GFP expression pattern in Col (**A**) and *bam1-3* (**B**) cotyledons at early stage of germination. From left to right, 1 dag, 1.5 dag, 2 dag. Bars = 50 μm. Signal intensities are coded from violet to red corresponding to increasing intensity levels (see color scale). (**C**) Mean ratio of DR5-GFP abundance at the margin cells versus other epidermal cells in Col and *bam1-3* at 1.5 dag. *n* = 8 with 20 cells for each plant were analyzed. Error bars represent SD, *p*-value was calculated according to Student’s *t*-test. *** denotes *p* < 0.001.

**Figure 4 ijms-26-04724-f004:**
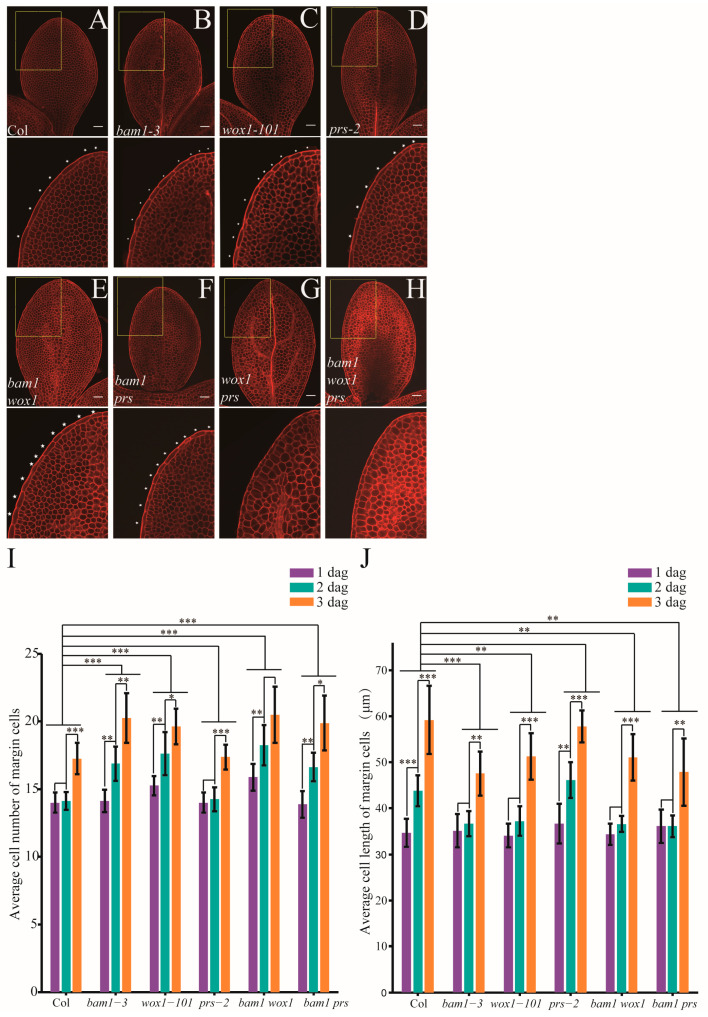
The role of WOX in regulating margin cell development. (**A**–**H**) mPS-PI-stained cotyledons in Col (**A**), *wox1 prs* (**B**), *bam1-3* (**C**), *wox1-101* (**D**), *prs-2* (**E**), *bam1 wox1* (**F**), *bam1 prs* (**G**) and *bam1 wox1 prs* (**H**) at 2 dag. Single slice of optimal sections of margin cells are shown in upper panels. Bottom panels show magnified images of upper panels (outlined in yellow boxes). White stars indicate margin cells. Bars = 50 μm. (**I**,**J**) Quantitative analysis of average cell number (**I**) and average cell length (**J**) of margin cells in Col and the mutants. Data are presented as mean ± SD. Student’s *t*-tests were performed to assess significance of differences between 2 dag to 1 dag and 3 dag to 2 dag in each group. The significance of differences between each mutant line and Col was assessed by two-way ANOVA. (*, ** and *** denote *p* < 0.05, *p* < 0.01 and *p* < 0.001, respectively. *n* = 8).

**Figure 5 ijms-26-04724-f005:**
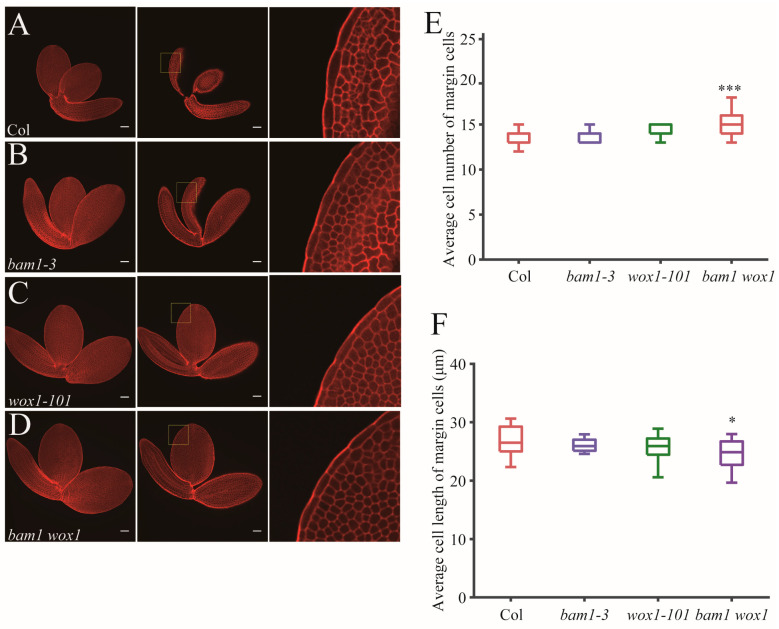
BAM1 and WOX1 synergistically regulate margin cell development in mature embryos. (**A**–**D**) mPS-PI-stained mature embryos of Col (**A**), *bam1-3* (**B**), *wox1-101* (**C**) and *bam1 wox1* (**D**). Maximum projection images of Z-stacks are shown in left panels. Single slices of optimal section of margin cells are shown in middle panels. Right panels show magnified images of middle panels (outlined in yellow boxes). Bars = 50 μm. (**E**,**F**) Quantitative analysis of average cell number (**E**) and average cell length (**F**) of margin cells in Col and the mutants. Data are presented as mean ± SD. Student’s *t*-tests were performed to assess significance of the differences between mutant lines and Col (* *p* < 0.05, *** *p* < 0.001, *n* = 14).

**Figure 6 ijms-26-04724-f006:**
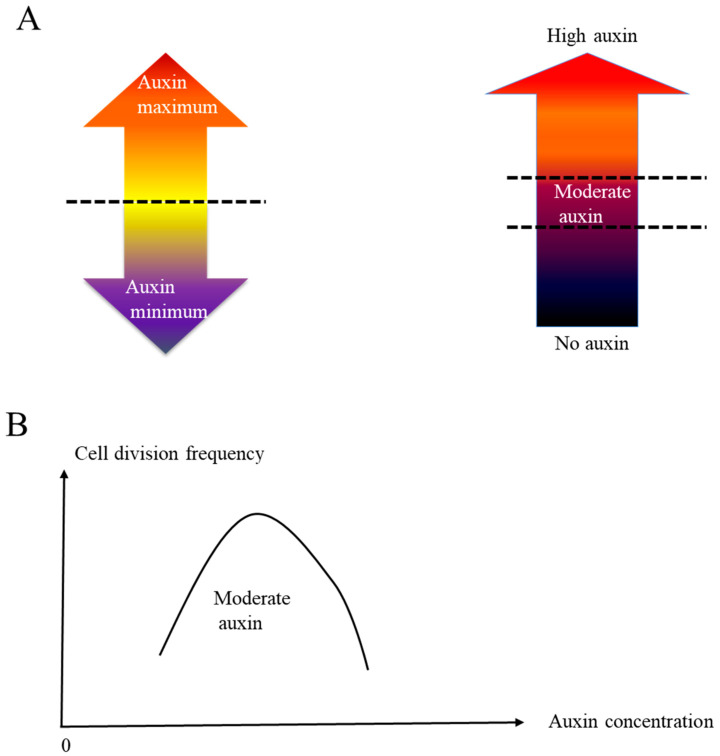
A moderate auxin concentration model for cell division control in margin cells. (**A**) Basic description of auxin minimum model (left panel) and moderate auxin concentration model (right panel). (**B**) The relationship of auxin concentration and cell division frequency. Our data suggest that there is a moderate auxin concentration range which promotes margin cell division. In normal conditions without exogenous auxin, this moderate auxin concentration showed up around 2 dag. However, if we treat the seedlings with low-concentration exogenous auxin at 1.5 dag, the moderate auxin concentration will be achieved earlier and additional cell division happens at this stage.

**Table 1 ijms-26-04724-t001:** Frequency (%) of cell division at different stages (*n* = 50, five repetitions).

Genotype	Stage	Transversal Cell Division	Longitudinal Cell Division
Col	1 dag	27.2 (±1.1)	15.2 (±2.3)
	1.5 dag	42.8 (±1.8)	28.4 (±1.7)
	2 dag	8.4 (±0.9)	14.4 (±1.7)
*bam1-3*	1 dag	42.6 (±2.5)	30.8 (±1.8)
	1.5 dag	38.2 (±3.5)	43.2 (±2.3)
	2 dag	15.6 (±2.2)	19.2 (±2.3)
*wox1-101*	1 dag	30 (±2.4)	35.2 (±3.0)
	1.5 dag	42.8 (±3.6)	36.4 (±3.0)
	2dag	10 (±2.4)	4.4 (±1.7)
*bam1 wox1*	1 dag	40.4 (±2.6)	26.8 (±2.3)
	1.5 dag	26.4 (±1.7)	32.4 (±4.3)
	2 dag	7.6 (±1.7)	6.8 (±2.3)

The values shown in the table are the mean ± standard deviation. There were five replicates.

**Table 2 ijms-26-04724-t002:** Frequency (%) of cell division with different concentration NAA treatment in Col at 1.5 dag (*n* = 30, five repetitions).

Chemical	Transversal Cell Division	Longitudinal Cell Division
EtOH	25.3 (±3.8)	24.0 (±6.0)
0.1 nM NAA	25.3 (±4.5)	28.7 (±1.8)
1 nM NAA	30.7 (±2.8)	42.7 (±2.8)
10 nM NAA	23.3 (±2.4)	26.7 (±4.1)
100 nM NAA	10.0 (±2.4)	11.3 (±3.8)

The values shown in the table are the mean ± standard deviation. There were five replicates.

**Table 3 ijms-26-04724-t003:** Frequency (%) of cell division with 1 nM NAA treatment at 1.5 dag (*n* = 30, five repetitions).

Genotype	Chemical	Transversal Cell Division	Longitudinal Cell Division
col	EtOH	34.0 (±6.0)	35.3 (±6.1)
	1 nM NAA	46.0 (±5.5)	43.3 (±9.4)
*bam 1-3*	EtOH	58.0 (±6.9)	51.3 (±12.4)
	1 nM NAA	39.3 (±10.4)	41.3 (±6.1)
*wox1-101*	EtOH	32.0 (±3.0)	41.3 (±6.9)
	1 nM NAA	42.7 (±5.5)	44.0 (±2.8)

The values shown in the table are the mean ± standard deviation. There were five replicates.

## Data Availability

Data is contained within the article and Appendix A.

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
