# Peer review of "Stem Cell Factors BAM1 and WOX1 Suppressing Longitudinal Cell Division of Margin Cells Evoked by Low-Concentration Auxin in Young Cotyledon of Arabidopsis"

_ijms, 2025, doi:10.3390/ijms26104724_

Round 1
Reviewer 1 Report
Comments and Suggestions for Authors
I could not review the manuscript due to the lack of Tables 2 & 3. Other major comments are listed below:
Table 1 (and probably Tables 2 & 3) seems to involve only one dataset. Given the large variation of this kind of analysis, the conclusion was not obtained from only one dataset. Considering that the experiment is not difficult, I ask authors to obtain at least 5 datasets and perform the statistical analysis to obtain the conclusion.
The method how to treat NAA is not described in the M&M section. Did authors spray NAA solution, immerse seedlings in the NAA solution, or mix NAA into the gel? The method should be described.
Related to the previous point, one of the punchlines of the manuscript is "division of margin cells evoked by low concentration auxin" as described in the title and abstract, and authors propose the "moderate auxin concentration model". However, actual auxin concentration change in cotyledon margin cells under exogenous NAA treatments is not analyzed and the conclusion and the model cannot be obtained from the data. To obtain such conclusion, authors should at least compare the auxin response without and with exogenous NAA treatment at 1 nM and 100 nM. This is not difficult because authors already have pDR5::GFP-ER line.
I could not understand how the description "BAM1 acts at a relatively later stage, from 1 dag to 2 dag" (line 210) was concluded from Fig. S2. In addition, it is obvious that BAM1 has a function in embryonic stage described in Fig. 5.
Authors stated "BAM1 seems to play roles in margin cell development in embryo stage with WOX1 which may be hindered by genetic redundancy with other BAMs/RLKs" (line 211). A part of this hypothesis is relatively easily confirmed because authors have bam1 bam2 bam3. I would like to ask authors to analyze the phenotype of bam1 bam2 bam3 to confirm the hypothesis.
Author Response
Thank you very much for taking the time to review this manuscript. Please find the detailed responses below and the corresponding revisions/corrections highlighted/in track changes in the re-submitted files.
Comments 1:Table 1 (and probably Tables 2 & 3) seems to involve only one dataset. Given the large variation of this kind of analysis, the conclusion was not obtained from only one dataset. Considering that the experiment is not difficult, I ask authors to obtain at least 5 datasets and perform the statistical analysis to obtain the conclusion.
Response 1: Thank you for pointing out my shortcomings. The five repeated experiments have been supplemented. Please refer to the Table section of the paper.
Comments 2:The method how to treat NAA is not described in the M&M section. Did authors spray NAA solution, immerse seedlings in the NAA solution, or mix NAA into the gel? The method should be described.
Response 2:We immerse seedlings in the NAA solution. Please refer to the method we added in part ‘4.6. NAA treatment '
Comments 3:Related to the previous point, one of the punchlines of the manuscript is "division of margin cells evoked by low concentration auxin" as described in the title and abstract, and authors propose the "moderate auxin concentration model". However, actual auxin concentration change in cotyledon margin cells under exogenous NAA treatments is not analyzed and the conclusion and the model cannot be obtained from the data. To obtain such conclusion, authors should at least compare the auxin response without and with exogenous NAA treatment at 1 nM and 100 nM. This is not difficult because authors already have pDR5::GFP-ER line.
Response 3: Thank you for pointing out the deficiencies in my thesis. Indeed, I should test the effect of exogenous NAA treatment on pDR5::GFP-ER line. However, a related project is already being carried out by another collaborator, so I'm not at liberty to disclose the experimental results. Thank you very much for your understanding.
Comments 4:I could not understand how the description "BAM1 acts at a relatively later stage, from 1 dag to 2 dag" (line 210) was concluded from Fig. S2. In addition, it is obvious that BAM1 has a function in embryonic stage described in Fig. 5
Response 4:WOX1 may acts earlier stage in the germination process, as we detected WOX1 promoter activity at 1 dag( Fig. S2D), but BAM1 promoter activity was not detected at this stage (Fig. S2A). bam1 mutant line show some phenotype as wide type line in embryonic stage(Fig. 5B and 5E),The relevant description can be found in the article from line 263 to line 268.
Comments 5:Authors stated "BAM1 seems to play roles in margin cell development in embryo stage with WOX1 which may be hindered by genetic redundancy with other BAMs/RLKs" (line 211). A part of this hypothesis is relatively easily confirmed because authors have bam1 bam2 bam3. I would like to ask authors to analyze the phenotype of bam1 bam2 bam3 to confirm the hypothesis.
Response 5 :Thank you for pointing out my shortcomings. We checked the phenotype of bam1 bam2 bam3 in embryonic stage,and it shows the some phenotype as bam1 mutant. However, other RLKs may have genetic redundancy with BAM1.
Please see the last part of the attached file.

Reviewer 2 Report
Comments and Suggestions for Authors
Line 19 of Abstract : How much is low concentration of auxin?
Line 22 of Abstract : moderate auxin concentration? How much?
In other parts you also talked about low and moderate concentrations of auxin, auxin maximum and minimum. Better to give a range of concentration to clarify them. For example for the Figure 6B you showed a curve based on auxin concentration but no unit is depicted and it is not clear what you do mean by auxin concentration (ppm, microgram, milligram, etc.).
Line 29: Give a definition of Pavement Cells for the readers with less knowledge about the topic.
In line 33 you mentioned Marginal cells. Are Marginal cells and Pavement cells same cells? Clarify it by giving definitions or explanations for both.
Line 68 What are BAM1 and BAM2? Put the whole name of gene at the first appearance as it is an important gene with more than 100 repeats in the text.
The Results section seems to be a part of Discussion. I understand that you wanted to explain the goal behind the obtained the results and how your results are relevant to previous studies in order to make it more interesting and attractive to readers. However, I recommend you to review Results and Discussion and then modify them when possible.
The Discussion section includes two paragraphs and the second one is 39 lines (250-289). I suggest that you divide the second paragraph into three or four paragraphs. Also, merge some parts of the Results section with the Discussion section.
In Materials and Methods: You mentioned that “The primers used for quantitative RT-PCR are shown in 309 Supplementary information, Table S1.” Which tool did you use to design the primers? If you used previously designed primers, give the reference.
Also you used different references for Materials and Methods and each time a reference was cited to say that one method has been used for this analysis without any more explanation. It was repeated for almost all parts of Materials and Methods (PCR, vector assembly, staining, labelling), thus, it can be a good idea to briefly explain the main points used.
Comments on the Quality of English LanguageAttention to English. Use the appropriate time tense and check the whole manuscript. For example in this sentence of Abstract: We use cotyledon margin cells from Arabidopsis plant as a new model system to investigate cell elongation and cell division during organ growth, and found that margin cells endured developmental phase transition 15 from the “elongation” phase to the “elongation and division” phase at the early stage in 16 germinating seedlings. You have used present time and then past tense.
Author Response
Thank you very much for taking the time to review this manuscript. Please find the detailed responses below and the corresponding revisions/corrections highlighted/in track changes in the re-submitted files
Comments 1: Line 19 of Abstract : How much is low concentration of auxin?
Response 1: Thank you for pointing out my shortcomings. 1 nM auxin is low concentration. The relevant expressions have been revised and marked in red in Line 19.
Comments 2:Line 22 of Abstract : moderate auxin concentration? How much?In other parts you also talked about low and moderate concentrations of auxin, auxin maximum and minimum. Better to give a range of concentration to clarify them. For example for the Figure 6B you showed a curve based on auxin concentration but no unit is depicted and it is not clear what you do mean by auxin concentration (ppm, microgram, milligram, etc.).
Response 2:Since the auxin concentration inside Arabidopsis cotyledons cannot be measured, I'm unable to tell you the exact value. The concentrations of exogenous auxin we applied were 0.1 nM, 1 nM, 10 nM, and 100 nM.
At the developmental stage of 1.5 dag, the endogenous auxin concentration within Arabidopsis thaliana cotyledons remains at a relatively low level, concurrently leading to the suppression of marginal cell division. When an exogenous application of auxin at a low concentration of 1 nM is administered at this precise developmental stage, it effectively elevates the auxin concentration within the cotyledons to an optimal range. This intervention subsequently triggers the premature activation of marginal cell division, deviating from the normal developmental trajectory. This is what we emphasized by the model of moderate auxin concentration, so it has no unit.
Comments 3:Line 29: Give a definition of Pavement Cells for the readers with less knowledge about the topic.In line 33 you mentioned Marginal cells. Are Marginal cells and Pavement cells same cells? Clarify it by giving definitions or explanations for both.
Response 3:Both pavement cells and marginal cells are epidermal cells of Arabidopsis, and they are mainly distinguished by their shapes and positions. Pavement cells are puzzle - shaped and located in the middle of the cotyledon surface. Marginal cells are rectangular and situated at the outermost part of the cotyledon. The description of the differences between the two is introduced from line 30 to line 33 and marked in red.
Comments 4:Line 68 What are BAM1 and BAM2? Put the whole name of gene at the first appearance as it is an important gene with more than 100 repeats in the text.
Response 4:Thank you for pointing out my shortcomings. the whole names of genes are added in line 17 and line 68.
Comments 5:The Results section seems to be a part of Discussion. I understand that you wanted to explain the goal behind the obtained the results and how your results are relevant to previous studies in order to make it more interesting and attractive to readers. However, I recommend you to review Results and Discussion and then modify them when possible. The Discussion section includes two paragraphs and the second one is 39 lines (250-289). I suggest that you divide the second paragraph into three or four paragraphs. Also, merge some parts of the Results section with the Discussion section.
Response 5:Thank you so much for your valuable suggestions. I have already made revisions according to your comments. Please refer to the "Discussion" section of the article for details.
Comments 6:In Materials and Methods: You mentioned that “The primers used for quantitative RT-PCR are shown in 309 Supplementary information, Table S1.” Which tool did you use to design the primers? If you used previously designed primers, give the reference.
Response 6:I used Primer Premier to design the primers. Information of primer sequences are shown in Table S1.
Comments 7:Also you used different references for Materials and Methods and each time a reference was cited to say that one method has been used for this analysis without any more explanation. It was repeated for almost all parts of Materials and Methods (PCR, vector assembly, staining, labelling), thus, it can be a good idea to briefly explain the main points used.
Response 7:I have enhanced the description of the experimental and methodology sections. Please refer to the Materials and Methods section of the paper.
Comments on the Quality of English Language
Comments 8:Attention to English. Use the appropriate time tense and check the whole manuscript. For example in this sentence of Abstract: We use cotyledon margin cells from Arabidopsis plant as a new model system to investigate cell elongation and cell division during organ growth, and found that margin cells endured developmental phase transition 15 from the “elongation” phase to the “elongation and division” phase at the early stage in 16 germinating seedlings. You have used present time and then past tense.
Response 8:Thank you so much for your valuable suggestions. I consistently used the past tense to describe the experiments and corrected the mistakes marked in yellow.

Round 2
Reviewer 1 Report
Comments and Suggestions for Authors
I review the manuscript written by Jiang et al. Authors found that the margin cells of cotyledons remain "elongation phase" until 2 dag and then shift to "elongation and division phase" in Col wild type. In bam1, wox1, and bam1 wox1 mutants, the cell divisions occurred earlier in 2 dag. In embryo, margin cell number slightly increased in bam1 wox1, suggesting the redundant role of BAM1 and WOX1 in the cell division in embryo. In bam1 mutant, auxin signal increased in margin cells compared with other cells. 1 nM NAA treatment increased the cell division of margin cells but 100 nM NAA treatment decreased it in the wild type. Authors emphasized the "moderate auxin concentration model" from the exogenous auxin treatment analysis, but they did not analyze internal auxin signaling under the treatment. This is critical weakness of the model. Since pDR5::GFP-ER is in their hand, it is not difficult to analyze the auxin signaling in the margin cells under 1 nM and 100 nM NAA treatment. However, authors stated that they cannot do the experiment due to unscientific reason. At least authors should describe the critical weakness of the model in Discussion as "The model was drawn mainly by exogenous NAA treatment and it is necessary to analyze the internal auxin signaling in the margin cells by 1 nM and 100 nM NAA treatment to conclude the model" or so. Other major comments are listed below:
- Authors did not supply the legend for supplemental figures, which is necessary to review the manuscript. I recommend authors to prepare the supplemental data files including supplemental figures and their legends.
- In the abstract, authors stated "This promotion of cell division is suppressed by BAM1 and WOX1." I think this is incorrect. More precisely, "This promotion of cell division did not occur in bam1 and wox1 mutants".
- Fig. 1C shows longitudinally quite short cells most likely produced by cell divisions at 3 dag. However, Fig. 1E does not seem to contain such short cells at 3 dag (if short cells are included, whisker should be extended downwards). Authors should explain this discrepancy.
- Authors described "Importantly, initiation of cell elongation of margin cells is retarded in bam1-3" (line 151). They cannot conclude this because cell length drastically decreases by cell division. Shorter cell length in bam1-2 at 2 dag may be caused by increased cell division not by retardation of cell elongation. Authors should describe both possibilities.
> Comments 4:I could not understand how the description "BAM1 acts at a relatively later stage,
> from 1 dag to 2 dag" (line 210) was concluded from Fig. S2. In addition, it is obvious that
> BAM1 has a function in embryonic stage described in Fig. 5
>
> Response 4:WOX1 may acts earlier stage in the germination process, as we detected WOX1
> promoter activity at 1 dag( Fig. S2D), but BAM1 promoter activity was not detected at this
> stage (Fig. S2A). bam1 mutant line show some phenotype as wide type line in embryonic
> stage(Fig. 5B and 5E),The relevant description can be found in the article from line 263 to
> line 268.
- Regarding above issue, it is obvious that BAM1 have a role in embryo as shown in Fig. 5. This is far stronger evidence than promoter reporter analysis. In the expression analysis such as promoter reporter analysis, no signals do not mean no expression but just under the detection limit. Promoter reporter analysis is sometimes unreliable as the case of AGAMOUS and FLC. I recommend authors change the description.
> Comments 5:Authors stated "BAM1 seems to play roles in margin cell development in embryo
> stage with WOX1 which may be hindered by genetic redundancy with other BAMs/RLKs" (line
> 211). A part of this hypothesis is relatively easily confirmed because authors have bam1 bam2
> bam3. I would like to ask authors to analyze the phenotype of bam1 bam2 bam3 to confirm the
> hypothesis.
>
> Response 5 :Thank you for pointing out my shortcomings. We checked the phenotype of bam1
> bam2 bam3 in embryonic stage,and it shows the some phenotype as bam1 mutant. Please see
> the picture here. However, other RLKs may have genetic redundancy with BAM1.
- Regarding above issue, thank you for supplying the figure. I recommend authors to put the figure as Supplementary Fig. 3, and revise the description as "BAM1 seems to play roles in margin cell development in embryo stage with WOX1 which may be hindered by genetic redundancy with other RLKs".
Minor point:
- Yuli Jiang appeared in the author list twice.
- Lines 210 - 216: Citations to Fig. 4 are mostly wrong. For example, "Figure 4A and 4B" (line 210) must be "Figure 4G". Carefully check all citations.
Author Response
Comments 1:I review the manuscript written by Jiang et al. Authors found that the margin cells of cotyledons remain "elongation phase" until 2 dag and then shift to "elongation and division phase" in Col wild type. In bam1, wox1, and bam1 wox1 mutants, the cell divisions occurred earlier in 2 dag. In embryo, margin cell number slightly increased in bam1 wox1, suggesting the redundant role of BAM1 and WOX1 in the cell division in embryo. In bam1 mutant, auxin signal increased in margin cells compared with other cells. 1 nM NAA treatment increased the cell division of margin cells but 100 nM NAA treatment decreased it in the wild type. Authors emphasized the "moderate auxin concentration model" from the exogenous auxin treatment analysis, but they did not analyze internal auxin signaling under the treatment. This is critical weakness of the model. Since pDR5::GFP-ER is in their hand, it is not difficult to analyze the auxin signaling in the margin cells under 1 nM and 100 nM NAA treatment. However, authors stated that they cannot do the experiment due to unscientific reason. At least authors should describe the critical weakness of the model in Discussion as "The model was drawn mainly by exogenous NAA treatment and it is necessary to analyze the internal auxin signaling in the margin cells by 1 nM and 100 nM NAA treatment to conclude the model" or so.
Response 1: Thank you for your suggestion once again. Yes, I truly need to include a discussion of the current deficiencies of my model in the discussion section. I have added this description to lines 338 to 340 of the article as you suggested.
Comments 2:Authors did not supply the legend for supplemental figures, which is necessary to review the manuscript. I recommend authors to prepare the supplemental data files including supplemental figures and their legends.
Response 2:Supplemental data files including supplemental figures and their legends have been supplied.
Comments 3:In the abstract, authors stated "This promotion of cell division is suppressed by BAM1 and WOX1." I think this is incorrect. More precisely, "This promotion of cell division did not occur in bam1 and wox1 mutants".
Response 3: Thank you for pointing out the deficiencies in my thesis. Yes, the original expression was too convoluted. I'd like to revise it to "This promotion of cell division did not occur in bam1 and wox1 mutants. " After making this modification, readers can understand my meaning more accurately.
Comments 4:Fig. 1C shows longitudinally quite short cells most likely produced by cell divisions at 3 dag. However, Fig. 1E does not seem to contain such short cells at 3 dag (if short cells are included, whisker should be extended downwards). Authors should explain this discrepancy.
Response 4:Yes, within a single cotyledon, some short cells are generated due to cell division, resulting in a large variation in cell length. However, what I statistically analyzed is the average length of the marginal cells of 16 cotyledons. The error bar demonstrates the differences in the average length of the marginal cells of these 16 cotyledons, which is why it is relatively small. To avoid such misunderstandings, I changed "cell length of margin cells" to "average cell length of margin cells" and "cell number of margin cells" to "average cell number of margin cells" in the figures of the article.
Comments 5:Authors described "Importantly, initiation of cell elongation of margin cells is retarded in bam1-3" (line 151). They cannot conclude this because cell length drastically decreases by cell division. Shorter cell length in bam1 at 2 dag may be caused by increased cell division not by retardation of cell elongation. Authors should describe both possibilities.
Response 5: Thank you for your excellent suggestion. Our conclusion was a bit arbitrary. Also, there is an EdU experiment at line 174 to verify the increased cell division in bam1-3. So, I have deleted this sentence.
Comments 6: Regarding above issue, it is obvious that BAM1 have a role in embryo as shown in Fig. 5. This is far stronger evidence than promoter reporter analysis. In the expression analysis such as promoter reporter analysis, no signals do not mean no expression but just under the detection limit. Promoter reporter analysis is sometimes unreliable as the case of AGAMOUS and FLC. I recommend authors change the description.
Response 6: Thank you for your suggestion. I changed the description as "WOX1 works from the mature embryo to 2dag while BAM1 may act at a relatively later stage" in line 271.
Comments 7: Regarding above issue, thank you for supplying the figure. I recommend authors to put the figure as Supplementary Fig. 3, and revise the description as "BAM1 seems to play roles in margin cell development in embryo stage with WOX1 which may be hindered by genetic redundancy with other RLKs".
Response 7:Thank you for your suggestion. I have added the phenotype of bam1 bam2 bam3 during the embryonic stage as Figure S3 in the supplementary materials. I revised the description as "BAM1 seems to play roles in margin cell development in embryo stage with WOX1 which may be hindered by genetic redundancy with other RLKs" in line 272 and line 273.
Comments 8:Yuli Jiang appeared in the author list twice
Response 8:I have corrected this error and marked an asterisk after Yuli Jiang to indicate that she is the corresponding author.
Comments 9:Lines 210 - 216: Citations to Fig. 4 are mostly wrong. For example, "Figure 4A and 4B" (line 210) must be "Figure 4G". Carefully check all citations.
Response 9:I have carefully checked all citations and revised these mistakes. The revisions are marked in yellow from line 212 to line 218.

Round 3
Reviewer 1 Report
Comments and Suggestions for Authors
All of my concerns have been solved.